# IN-CONTEXT LEARNING WITH ITERATIVE DEMONSTRATION SELECTION

## ABSTRACT

Spurred by advancements in scale, large language models (LLMs) have demonstrated strong few-shot learning ability via in-context learning (ICL). However, the performance of ICL has been shown to be highly sensitive to the selection of few-shot demonstrations. Selecting the most suitable examples as context remains an ongoing challenge and an open problem. Existing literature has highlighted the importance of selecting examples that are diverse or semantically similar to the test sample while ignoring the fact that the optimal selection dimension, *i.e.,* diversity or similarity, is task-specific. Leveraging the merits of both dimensions, we propose Iterative Demonstration Selection (IDS). Using zero-shot chain-of-thought reasoning (Zero-shot-CoT), IDS iteratively selects examples that are diverse but still strongly correlated with the test sample as ICL demonstrations. Specifically, IDS applies Zero-shot-CoT to the test sample before demonstration selection. The output reasoning path is then used to choose demonstrations that are prepended to the test sample for inference. The generated answer is followed by its corresponding reasoning path for extracting a new set of demonstrations in the next iteration. After several iterations, IDS adopts majority voting to obtain the final result. Through extensive experiments on tasks including reasoning, question answering, topic classification, and sentiment analysis, we demonstrate that IDS can consistently outperform existing ICL demonstration selection methods.

## 1 INTRODUCTION

With the recent advancements in scaling up model parameters, large language models (LLMs) showcase promising results on a variety of few-shot tasks through in-context learning (ICL), where the model is expected to directly generate the output of the test sample without updating parameters. This is achieved by conditioning on a manually designed prompt consisting of an optional task description and a few demonstration examples (Brown et al., 2020). Figure 1 shows an example describing how LLMs perform ICL on the sentiment analysis task. Given a few review-sentiment pairs as demonstrations, ICL combines them with the test sample as input, to the LLM for inference. The output, *i.e.,* 'Positive', is generated by the model autoregressively without any parameter updates.

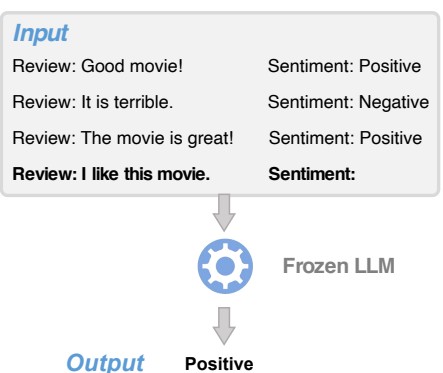

Figure 1: Illustration of in-context learning (ICL) on sentiment analysis. A frozen large language model directly generates the sentiment 'Positive' for the test sample 'I like this movie.' by taking the demonstrations and the test sample as input.

Despite the effectiveness, the performance of ICL has been shown to be highly sensitive to the selection of demonstration examples (Zhao et al., 2021). Different sets of demonstrations can yield performance ranging from nearly random to comparable with state-of-the-art models (Gao et al., 2021; Lu et al., 2022). To alleviate the above issue, researchers in ICL have proposed a number of methods to select a set of examples as few-shot demonstrations (Rubin et al., 2022; Liu et al., 2022; Li & Qiu, 2023; Wang et al., 2023b; Li et al., 2023a; Ma et al., 2023; An et al., 2023b). Nevertheless, most of the existing approaches are only

applicable to small language models as they typically require accessing model parameters or detailed output distributions which are usually not available for LLMs (Sun et al., 2022). Therefore, it is still a common practice to randomly select examples or select examples that are semantically similar to the test sample as demonstrations for LLMs, *i.e.,* considering diversity or similarity. While several approaches investigate the combination of similarity and diversity when prompting with explanations or exploring compositional generalization (Ye et al., 2022; An et al., 2023a), it remains unclear to us how to determine and leverage the optimal dimension for different tasks in ICL.

Actually, the optimal dimension for selecting demonstration examples is task-specific. As we will show in Section 4, the diversity dimension is superior to the similarity dimension on CommonsenseQA (Talmor et al., 2019) while the similarity dimension outperforms the diversity dimension on AGNews (Zhang et al., 2015). Thus, it is unreasonable to claim that one dimension is consistently better than the other across different tasks.

To fully leverage the merits of both dimensions, we propose Iterative Demonstration Selection (IDS) for ICL (Figure 2). IDS can iteratively select demonstration examples that are diverse but still have a strong correlation with the test sample through zero-shot chain-of-thought reasoning (Zero-shot-CoT) (Kojima et al., 2022). Specifically, Zero-shot-CoT, *e.g.,* "Let's think step by step.", is first applied to the test sample before selecting demonstrations to obtain a reasoning path. The training examples that are most semantically similar to the generated reasoning path are then selected as demonstrations. They are prepended to the test sample for inference. Note that IDS ensures that the generated answer is accompanied by the reasoning path through designed prompts. The new reasoning path is then used for extracting another set of demonstration examples by semantic similarity in the next iteration. After a few iterations, IDS adopts majority voting to obtain the final result. Empirical results on tasks spanning commonsense reasoning, question answering, topic classification, and sentiment analysis show that IDS can consistently outperform previous ICL demonstration selection baselines. In summary, our main contributions are:

- We consider both the diversity and similarity dimensions of ICL demonstration selection for LLMs. We identify that the optimal dimension for selecting demonstrations is task-specific and propose Iterative Demonstration Selection (IDS) to fully leverage the merits of both dimensions.
- With extensive experiments and analysis, we demonstrate the effectiveness of IDS on a variety of tasks. Our code base is available at `<redacted>`.

## 2 RELATED WORK

This work mainly explores how to select few-shot in-context learning demonstrations for LLMs by leveraging Zero-shot-CoT. In light of this, we review four lines of research that form the basis of this work: few-shot learning, in-context learning basics, demonstration selection for in-context learning, and chain-of-thought reasoning.

### 2.1 FEW-SHOT LEARNING

Few-shot learning aims to learn tasks with only a few labeled samples, which results in a big challenge,*i.e.,* over-fitting, for models as they typically require large amounts of data for training. Prior methods to address over-fitting mainly focused on augmenting the few-shot data (Gao et al., 2020; Qin & Joty, 2022), reducing the hypothesis space (Triantafillou et al., 2017; Hu et al., 2018), or optimizing the strategy for searching the best hypothesis (Ravi & Larochelle, 2017; Finn et al., 2017). More recently, LLMs have demonstrated strong few-shot learning ability through in-context learning without any parameter updates (Brown et al., 2020).

### 2.2 IN-CONTEXT LEARNING

Brown et al. (2020) first showed that a frozen GPT-3 model can achieve impressive results on a variety of few-shot NLP tasks through conditioning on manually designed prompts consisting of task descriptions and several demonstration examples. Since then many efforts have been made on in-context learning (ICL). Chen et al. (2022); Min et al. (2022a); Wei et al. (2023a) demonstrated that the ICL ability of language models can be further improved through self-supervised or supervised training. Some analytical studies attempted to understand what factors affect ICL performance (Zhao

et al., 2021; Shin et al., 2022; Wei et al., 2022a; Min et al., 2022b; Yoo et al., 2022; Wei et al., 2023b) and why ICL works (Xie et al., 2022; Olsson et al., 2022; Li et al., 2023b; Pan, 2023; Dai et al., 2023). Other ongoing research on ICL has also explored (*i*) demonstration designing, including demonstration selection (Liu et al., 2022; Rubin et al., 2022; Wang et al., 2023b), demonstration ordering (Lu et al., 2022), and demonstration formatting (Wei et al., 2022c; Wang et al., 2022c; Zhou et al., 2023; Zhang et al., 2023b), (*ii*) applications of ICL (Ding et al., 2022; Meade et al., 2023; Zheng et al., 2023), and (*iii*) ICL beyond text (Wang et al., 2023c; Huang et al., 2023; Zhu et al., 2023; Wang et al., 2023a).

### 2.3 Demonstration Selection for In-context Learning

The performance of ICL has been shown to be highly sensitive to the selection of demonstration examples (Zhao et al., 2021). Existing methods to solve this problem can be mainly divided into two categories. First, *unsupervised* methods rely on pre-defined metrics. Liu et al. (2022) proposed to select the closest neighbors as demonstrations. In contrast, Levy et al. (2022) selected diverse demonstrations to improve in-context compositional generalization. More recent studies have explored leveraging the output distributions of language models to select few-shot demonstrations (Wu et al., 2022; Nguyen & Wong, 2023; Li & Qiu, 2023). Second, *supervised* methods involve model training. Rubin et al. (2022); Ye et al. (2023); Li et al. (2023a); Luo et al. (2023) proposed to learn to retrieve demonstration examples. Wang et al. (2023b) posited LMs as implicit topic models to facilitate demonstration selection. In addition, some studies (Zhang et al., 2022; Scarlatos & Lan, 2023) attempted to select demonstrations based on reinforcement learning. However, most of the existing methods are not applicable to LLMs as model parameters or output distributions are typically not available for LLMs (Sun et al., 2022), which motivates us to propose our simple but effective approach (IDS).

### 2.4 Chain-of-Thought Reasoning

Chain-of-thought (CoT) reasoning induces LLMs to produce intermediate reasoning steps before generating the final answer (Wei et al., 2022b). Depending on whether there are manually designed demonstrations, current CoT reasoning methods mainly include Manual-CoT and Zero-shot-CoT. In Manual-CoT, human-labeled reasoning paths are used to perform CoT reasoning (Wei et al., 2022b; Zhou et al., 2022; Wang et al., 2022a; Li et al., 2022; Wang et al., 2022b). In contrast, LLMs leverage self-generated rationales for reasoning in Zero-shot-CoT (Kojima et al., 2022; Zelikman et al., 2022; Zhang et al., 2023a; Diao et al., 2023). The ongoing research on CoT reasoning has also explored (*i*) multimodal reasoning (Zhang et al., 2023c; Wu et al., 2023), (*ii*) distilling knowledge from LLMs (Ho et al., 2022; Fu et al., 2023), and (*iii*) iterative optimization (Shinn et al., 2023; Madaan et al., 2023; Paul et al., 2023).

## 3 Problem Formulation

Given the test set $\mathcal{D}_{\text{test}}$ and the training set $\mathcal{D}_{\text{train}}$, the goal of ICL demonstration selection is to find an optimal subset $\mathcal{S} = \{(x_1, y_1), ..., (x_k, y_k)\}$ ($k$-shot) of $\mathcal{D}_{\text{train}}$ as demonstration examples for each test sample $(\hat{x}_i, \hat{y}_i)$ to maximize the overall task performance on $\mathcal{D}_{\text{test}}$. More formally, the optimal selection method $\tilde{h}$ is defined as:

$$\tilde{h} = \arg\max_{h \in \mathcal{H}} \sum_{i=1}^{|\mathcal{D}_{\text{test}}|} \delta_{\text{LLM}([h(\mathcal{D}_{\text{train}}, \hat{x}_i, \hat{y}_i), \hat{x}_i]), \hat{y}_i} \tag{1}$$

where $\mathcal{H}$ is the hypothesis space for searching demonstration examples, $h(\mathcal{D}_{\text{train}}, \hat{x}_i, \hat{y}_i)$ refers to demonstrations selected for $(\hat{x}_i, \hat{y}_i)$ using $h$, $[,]$ stands for concatenation, and $\delta_{a,b}$ is the Kronecker delta function: $\delta_{a,b} = 1$ if $a$ equals $b$, otherwise $\delta_{a,b} = 0$. In this work, we aim to find the optimal method $\tilde{h}$ by leveraging Zero-shot-CoT.

Table 1: Results of different methods on CommonsenseQA, BoolQ, AGNews and SST2. The optimal dimension for selecting ICL demonstrations is task-specific.

| | CommonsenseQA | BoolQ | AGNews | SST2 |
|---|---|---|---|---|
| Similar-ICL-Consistency (**Similarity**) | 76.0 | **84.8** | **90.0** | 94.3 |
| Random-ICL-Voting (**Diversity**) | **79.0** | 83.5 | 88.0 | **95.2** |

## 4    WHAT MAKES GOOD IN-CONTEXT DEMONSTRATIONS?

As demonstrated in previous work (Zhao et al., 2021), the overall task performance is highly sensitive to the selection method $h$. Different sets of demonstration examples can yield significantly different performance. For example, Zhang et al. (2022) shows that the minimum and maximum ICL performance due to random sampling differs by $> 30\%$ on 4 classification tasks, which emphasizes the importance of selecting good demonstrations for LLMs.

A natural question is: what makes good in-context demonstrations? For LLMs, it is still a common practice to select a subset $\mathcal{S}$ consisting of examples that are diverse or semantically similar to the test sample as demonstrations, *i.e.,* considering the diversity or similarity of $\mathcal{S}$. To investigate whether one dimension is consistently better than the other one across different tasks, we conduct some pilot experiments on CommonsenseQA (Talmor et al., 2019), BoolQ (Clark et al., 2019), AGNews (Zhang et al., 2015) and SST2 (Socher et al., 2013). Specifically, we randomly sample 100 examples from the original test set for experiments and conduct 4-shot learning using GPT-3.5 (gpt-3.5-turbo).

Following Zhang et al. (2023a), we use Sentence-BERT (Reimers & Gurevych, 2019) to encode all samples. For each test sample, the Similar-ICL method selects the top-4 similar training data based on cosine similarity while the Random-ICL method randomly samples 4 training examples as few-shot demonstrations. Inspired by Wang et al. (2022a), we apply *self-consistency* with 3 decoding paths (temperature 0.7) to Similar-ICL (named **Similar-ICL-Consistency**) and run Random-ICL 3 times before majority voting (named **Random-ICL-Voting**) to improve the robustness.

The results of different methods on four datasets are reported in Table 1. We can observe that the diversity dimension outperforms the similarity dimension on CommonsenseQA and SST2 while the similarity dimension is superior to the diversity dimension on BoolQ and AGNews. Therefore, the optimal dimension for selecting demonstration examples is task-specific. Thus, it is unreasonable to claim that one dimension is consistently better than the other one in ICL demonstration selection.

Intuitively, semantically similar examples can help the model correctly answer the test query as they might share similar input-output patterns with the test sample which could unleash GPT-3.5's power of text generation. To further understand why the similarity dimension underperforms the diversity dimension on CommonsenseQA, we present a case study in Table 2. We can see that the answer of the final demonstration example extracted by Similar-ICL-Consistency, *i.e.,* 'most buildings' is also in the options list of the test sample, which misleads the decision process of the model, leading to a wrong answer. In addition, the selected demonstrations might not include enough important information as high similarity also results in redundancy.

Considering the strengths and weaknesses of both dimensions, we aim to design a method that can select demonstration examples that are diverse (minimizing misleading information) but still strongly correlated with the test sample, which is introduced in the next section.

## 5    ITERATIVE DEMONSTRATION SELECTION

Based on the observations and considerations in Section 4, we introduce Iterative Demonstration Selection (IDS) for ICL demonstration selection (see Figure 2 for an illustration), which can fully leverage the merits of both dimensions, *i.e.,* diversity and similarity. Intuitively, the demonstrations that are similar to the *reason* for answering a sample are strongly correlated with this sample. Therefore, we propose to incorporate zero-shot chain-of-thought reasoning (Zero-shot-CoT) into IDS to iteratively select demonstration examples that are diverse but still have a strong correlation with the test sample.

Table 2: Examples of Similar-ICL-Consistency (first decoding path) and Random-ICL-Voting (first run) for constructing demonstration examples. The upper part is the input to LLMs, including few-shot demonstrations, and the lower part is the predicted answer. Similar-ICL-Consistency gives the wrong answer 'most buildings' which is actually the output of the final demonstration example, indicating that the decision process of the model is misled by this similar sample.

| Similar-ICL-Consistency | Random-ICL-Voting |
|---|---|
| Which choice is the correct answer to the question? | Which choice is the correct answer to the question? |
| **Examples**:
**Question**: If you have cleaned off dust here it may be difficult to do your homework where? Answer Choices: (A) desktop (B) closet (C) most buildings (D) surface of earth (E) stove
**Answer**: A
**Question**: Where is dust likely to be under? Answer Choices: (A) closet (B) ground (C) windowsill (D) attic (E) carpet
**Answer**: E
**Question**: Where would you find a dustbin that is being used? Answer Choices: (A) utility closet (B) ground (C) cupboard (D) broom closet (E) kitchen
**Answer**: E
**Question**: Dust accumulates where? Answer Choices: (A) ceiling (B) library (C) surface of earth (D) **most buildings** (E) desktop
**Answer**: D | **Examples**:
**Question**: She had a busy schedule, she had to run errands and pick up the kids the second she did what? Answer Choices: (A) make time for (B) take money (C) go outdoors (D) leave work (E) field
**Answer**: D
**Question**: What is the worst outcome of an injury? Answer Choices: (A) cause death (B) cause bleeding (C) falling down (D) become infected (E) claim insurance
**Answer**: A
**Question**: Mom said that Sarah should stay in bed until she was able to go to school again.. What did mom say to Sarah when she tried to get up? Answer Choices: (A) you're sick (B) were sick (C) more rest (D) rest more (E) get back under the covers
**Answer**: A
**Question**: John got a raise, but he lost rank. Overall, it was a good what? Answer Choices: (A) demotion (B) push down (C) go off strike (D) lower (E) go off strike
**Answer**: A |
| The response should follow the format: Answer: {A, B, C, D or E}
Here is the test data.
**Question**: John wanted to clean all of the dust out of his place before settling down to watch his favorite shows. What might he hardest do dust? Answer Choices: (A) closet (B) under the bed (C) television (D) attic (E) **most buildings** | The response should follow the format: Answer: {A, B, C, D or E}
Here is the test data.
**Question**: John wanted to clean all of the dust out of his place before settling down to watch his favorite shows. What might he hardest do dust? Answer Choices: (A) closet (B) under the bed (C) television (D) attic (E) most buildings |
| **Answer**: E ✗ | **Answer**: D ✓ |

Specifically, for each test sample $(\hat{x}_i, \hat{y}_i)$, IDS mainly consists of four steps:

1. We apply **Zero-shot-CoT**, *i.e.,* "Let's think step by step." to the test sample $(\hat{x}_i, \hat{y}_i)$ before selecting demonstrations to obtain a reasoning path $\hat{R}$.

2. The **reasoning path** $\hat{R}$ is then used to select top-$k$ ($k$ is the number of shot) most semantically similar training examples $\{(x_1, y_1), ..., (x_k, y_k)\}$ as few-shot demonstrations. We use Sentence-BERT (Reimers & Gurevych, 2019) to encode the reasoning path $\hat{R}$ and training examples to obtain the contextual representations and use cosine similarity to measure the similarity between two representations.

3. The selected $k$ training examples $\{(x_1, y_1), ..., (x_k, y_k)\}$ are then prepended to the test sample $(\hat{x}_i, \hat{y}_i)$ for ICL. During inference, we ensure that the generated answer $\hat{A}$ is accompanied by its corresponding reasoning path $\hat{R}$ through designed prompts, *e.g.,* "The response should follow the format: Sentiment: {positive or negative}\nReason: {reason}". Note that **Zero-shot-CoT** is also applied in this step to improve the quality of generated reasoning paths. After ICL, we go back to Step 2 for *iterations* using the *new* reasoning path $\hat{R}$.

4. After $q$ rounds of iterations between Step 2 and 3, we adopt **majority voting** on all $\hat{A}$ to obtain the final result $\hat{A}_{final}$.

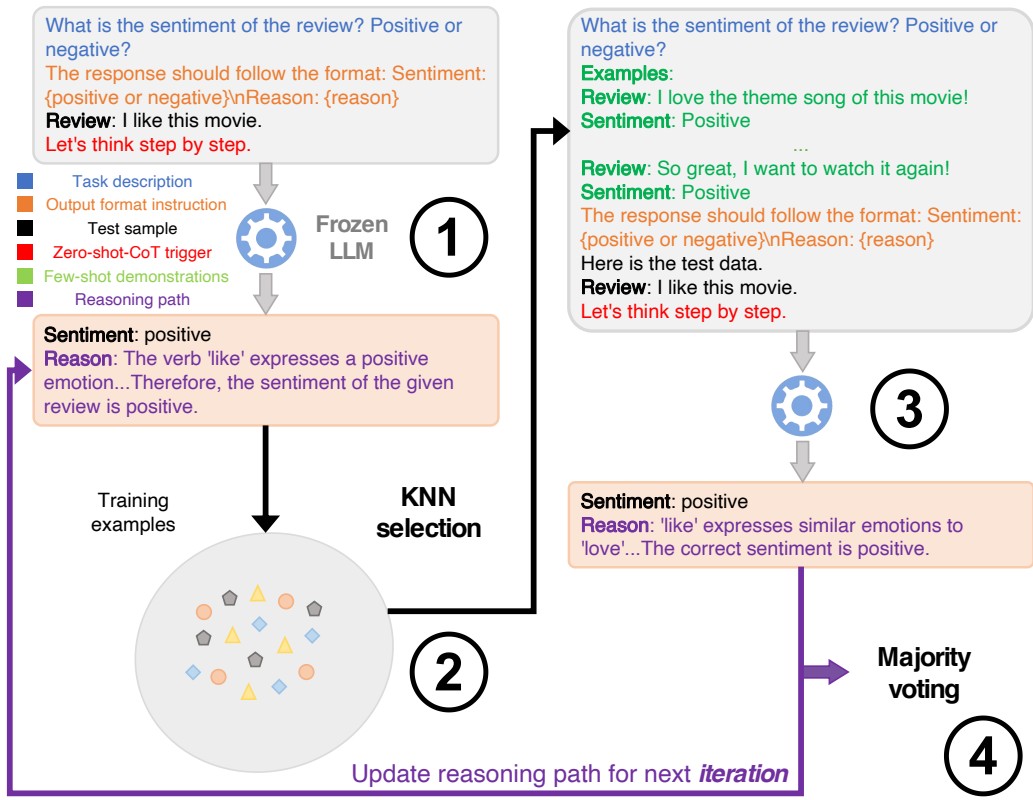

Figure 2: Illustration of our proposed Iterative Demonstration Selection (IDS). IDS first applies Zero-shot-CoT to the test sample to obtain a reasoning path, which is then used to select few-shot demonstrations from training examples through KNN. The selected demonstration examples are prepended to the test sample for ICL. To obtain the new reasoning path for extracting another set of demonstrations in the next iteration, an instruction for output format is inserted before the test sample. After several iterations, IDS uses majority voting to obtain the final result.

Obviously, the selected demonstration examples are strongly correlated with the original test sample, *i.e.,* achieving similarity, as they are selected by the generated reasoning paths. And they can be different during iterations to ensure diversity because the reasoning paths vary in different iterations. Note that there is *no* reasoning path in few-shot demonstrations (as shown in the green dashed area in Figure 2). The reasoning path only exists in the output of LLMs.

In addition, we show the instructions and input formats of different types of tasks for ICL in Appendix A.1 and illustrate the algorithm for the whole selection process in Appendix A.2.

## 6 EXPERIMENTS

In this section, we first describe the tasks and datasets, and then introduce methods compared in our work. Finally, we present the experimental results.

### 6.1 EXPERIMENTAL SETUP

**Tasks and Datasets** We mainly investigate 6 different datasets covering 4 representative task categories: commonsense reasoning (CommonsenseQA (Talmor et al., 2019)), question answering (BoolQ (Clark et al., 2019)), topic classification (AGNews (Zhang et al., 2015) and DBPedia (Lehmann et al., 2015)) and sentiment analysis (SST2 (Socher et al., 2013) and Amazon Review (McAuley et al., 2015)). For each dataset, we randomly sample at most 10000 examples from the original training set as $\mathcal{D}_{train}$ and at most 2000 test examples as $\mathcal{D}_{test}$ for evaluating the

Table 3: Accuracy (%) of different methods on 6 datasets. **Bold** indicates the best result. IDS is consistently better than all previous baselines.

| Method | CommonsenseQA | BoolQ | AGNews | DBPedia | SST2 | Amazon | Average |
|---|---|---|---|---|---|---|---|
| Top-$k$-Consistency | $73.7_{\pm0.1}$ | $83.6_{\pm0.4}$ | $88.1_{\pm0.8}$ | $97.4_{\pm0.2}$ | $93.6_{\pm0.4}$ | $91.4_{\pm1.1}$ | $88.0_{\pm0.2}$ |
| Random-Voting | $76.4_{\pm0.7}$ | $82.2_{\pm0.7}$ | $85.4_{\pm1.5}$ | $96.2_{\pm0.3}$ | $94.4_{\pm0.2}$ | $91.5_{\pm0.7}$ | $87.7_{\pm0.3}$ |
| Cluster-Voting | $75.4_{\pm0.8}$ | $82.9_{\pm0.6}$ | $86.0_{\pm1.9}$ | $95.8_{\pm0.5}$ | $93.3_{\pm1.1}$ | $89.7_{\pm1.9}$ | $87.2_{\pm0.5}$ |
| Top-$k$-Consistency-CoT | $74.5_{\pm0.2}$ | $87.1_{\pm0.2}$ | $89.3_{\pm0.8}$ | $97.6_{\pm0.5}$ | $95.2_{\pm0.4}$ | $95.3_{\pm0.7}$ | $89.8_{\pm0.1}$ |
| Random-Voting-CoT | $77.0_{\pm0.2}$ | $87.3_{\pm0.6}$ | $87.0_{\pm1.6}$ | $96.6_{\pm0.4}$ | $95.6_{\pm0.1}$ | $95.5_{\pm0.5}$ | $89.8_{\pm0.2}$ |
| Cluster-Voting-CoT | $76.5_{\pm0.3}$ | $86.4_{\pm0.7}$ | $86.8_{\pm1.2}$ | $95.9_{\pm1.0}$ | $95.2_{\pm0.4}$ | $95.2_{\pm0.7}$ | $89.3_{\pm0.1}$ |
| G-fair-Prompting | $75.5_{\pm0.3}$ | $84.8_{\pm0.7}$ | $88.9_{\pm1.0}$ | $97.0_{\pm0.5}$ | $94.6_{\pm0.3}$ | $94.4_{\pm0.8}$ | $89.2_{\pm0.2}$ |
| Skill-KNN | $75.2_{\pm0.2}$ | $85.9_{\pm0.5}$ | $88.7_{\pm0.9}$ | $96.9_{\pm0.6}$ | $94.9_{\pm0.2}$ | $95.0_{\pm0.7}$ | $89.4_{\pm0.1}$ |
| IDS | $\mathbf{78.1}_{\pm0.1}$ | $\mathbf{87.8}_{\pm0.8}$ | $\mathbf{89.8}_{\pm0.8}$ | $\mathbf{97.9}_{\pm0.4}$ | $\mathbf{95.8}_{\pm0.2}$ | $\mathbf{95.7}_{\pm0.5}$ | $\mathbf{90.9}_{\pm0.1}$ |

performance of selected demonstrations. The detailed information of different datasets is shown in Appendix A.3. To reduce the randomness, we run every experiment five times with different random seeds (resulting in different training and test samples if not using the whole set) and report the average results. Without specification, we use $k = 4$ number of demonstrations following Wang et al. (2023b) and set the number of iterations $q$ to 3.

**Methods Compared**   We mainly use GPT-3.5 (gpt-3.5-turbo) as the LLM and compare our IDS with the following methods in the experiments for selecting ICL demonstrations:

- **Top-$k$-Consistency** (Liu et al., 2022) selects the *top-k* semantically similar examples from the training set $\mathcal{D}_{\text{train}}$ as demonstrations for each test sample and applies *self-consistency* (Wang et al., 2022a) with $q$ decoding paths (temperature 0.7) to match the number of iterations. Following Zhang et al. (2023a), all samples are encoded by Sentence-BERT (Reimers & Gurevych, 2019) to obtain contextual representations for calculating the cosine similarity.

- **Random-Voting** randomly selects $k$ examples from $\mathcal{D}_{\text{train}}$ as few-shot demonstrations for every test sample and runs experiments $q$ times before majority voting.

- **Cluster-Voting** partitions $\mathcal{D}_{\text{train}}$ into $k$ clusters and selects a representative example from each cluster to form demonstrations. Following Zhang et al. (2023a), we choose the sample closest to the centroid in each cluster as the representative example. Same as Random-Voting, after running experiments $q$ times, Cluster-Voting adopts majority voting to obtain the final result.

As mentioned in Section 5, IDS ensures that the generated answer is followed by the reasoning path during ICL inference, which might influence the performance of ICL. To ensure a fair comparison with baselines, we apply the same prompt, *e.g.,* "The response should follow the format: Sentiment: {positive or negative}\nReason: {reason}", and *Zero-shot-CoT* to baseline methods to allow them simultaneously generate answers and reasoning paths, resulting in new variants of the three baselines: **Top-$k$-Consistency-CoT**, **Random-Voting-CoT** and **Cluster-Voting-CoT**. Besides, we also compare IDS with two latest ICL demonstration selection approaches: G-fair-Prompting (Ma et al., 2023) and Skill-KNN (An et al., 2023b).

## 6.2   MAIN RESULTS

Table 3 shows the average performance scores of different methods on all investigated datasets. From the results, we can observe that

- Our proposed IDS consistently outperforms previous baselines on all datasets with a negligible increase in API request cost (Zero-shot-CoT in the first step), which demonstrates that our method can indeed effectively and efficiently select better ICL demonstration examples. On average, IDS yields about 1.1% performance boost compared to the best baseline as it can fully leverage the merits of both selection dimensions (diversity and similarity). In particular, IDS outperforms Top-$k$-Consistency-CoT by 3.6% on CommonsenseQA and Random-Voting-CoT by 2.8% on AGNews.

- During ICL, simultaneously generating answers and reasoning paths can improve the performance on all datasets even if they are not reasoning tasks in the conventional sense, *e.g.,* sentiment analysis. Specifically, all three variants outperform the corresponding baseline by about 2% on average.

- Cluster-Voting (or its variant) underperforms Top-$k$-Consistency and Random-Voting (or their variants) on most datasets, which is inconsistent with the conclusion in AutoCoT (Zhang et al., 2023a). As shown in Zhang et al. (2023a), selecting a representative sample from each cluster and generating the corresponding reasoning chain using Zero-shot-CoT to construct chain-of-thought demonstrations can achieve better performance than selection by similarity or random selection. We speculate that this is because there is no rationale in ICL demonstration examples, which eliminates the advantage of cluster-based methods in mitigating misleading caused by rationale errors. In addition, Cluster-Voting (or its variant) selects demonstrations at the dataset level, *i.e.,* all test samples use the same demonstration examples, which is not as flexible as other instance-level methods.

## 6.3 ANALYSIS

**Different Numbers of Demonstrations** While we use $k = 4$ number of demonstrations for all experiments, we also evaluate the effectiveness of IDS with different $k$. We randomly choose one seed for experiments and report the average results of the 6 datasets in Table 4. We can see that IDS consistently outperforms Top-$k$-Consistency-

Table 4: Accuracy (%) of Top-$k$-Consistency-CoT and IDS with different numbers of demonstrations $k$.

|  | 2 | 4 | 6 | 8 |
|---|---|---|---|---|
| Top-$k$-Consistency-CoT | 89.6 | 90.0 | 90.0 | 90.1 |
| IDS | **90.4** | **90.9** | **90.7** | **90.6** |

CoT with different numbers of demonstrations. In addition, more demonstration examples do **not** guarantee better ICL performance, which is consistent with the observation in Wang et al. (2023b).

**Different Numbers of Iterations** We mainly use $q = 3$ number of iterations for all experiments. To verify whether the performance gain of IDS is consistent across different numbers of iterations, we conduct controlled experiments with $q = \{1, 5, 7\}$. The average results of the 6 datasets with a randomly selected seed are reported in Table 5. IDS consistently outperforms Top-

Table 5: Accuracy (%) of Top-$k$-Consistency-CoT and IDS with different numbers of reasoning paths or iterations.

|  | 1 | 3 | 5 | 7 |
|---|---|---|---|---|
| Top-$k$-Consistency-CoT | 89.8 | 90.0 | 90.1 | 90.0 |
| IDS | **90.4** | **90.9** | **90.8** | **90.8** |

$k$-Consistency-CoT with different $q$. Interestingly, the performance of ICL does not always improve with the number of iterations, which might be because increased iterations can also lead to unnecessary noise.

**More Complex Tasks** To better demonstrate the effectiveness of IDS, we further conduct experiments on two more complex datasets: GSM8K (Cobbe et al., 2021) (mathematical reasoning) and LogiQA (Liu et al., 2020) (logical reasoning). Specifically, we randomly sample 500 test examples for experiments and report the results in Table 6. IDS brings an average relative improvement of about 3%, demonstrating its superiority over baselines.

Table 6: Accuracy (%) of Top-$k$-Consistency-CoT and IDS on GSM8K and LogiQA.

|  | GSM8K | LogiQA |
|---|---|---|
| Top-$k$-Consistency-CoT | 76.2 | 45.4 |
| IDS | **78.4** | **46.8** |

**Robustness to Model Types** To demonstrate the robustness of IDS to model types, we conduct controlled experiments with GPT-4 (gpt-4). Specifically, we randomly select one seed and sample 200 test examples per dataset for experiments due to the expensive cost. From the average results reported in Table 7, we can observe that IDS still achieves better performance than Top-$k$-Consistency-CoT when using GPT-4 as the LLM, showing its robustness to different LLMs.

Table 7: Accuracy (%) of Top-$k$-Consistency-CoT and IDS with different LLMs (gpt-3.5-turbo and gpt-4). For gpt-4, we randomly sample 200 test examples per dataset.

|  | gpt-3.5-turbo | gpt-4 |
|---|---|---|
| Top-$k$-Consistency-CoT | 90.0 | 92.8 |
| IDS | **90.9** | **93.6** |

**Robustness to Embedding Model** Instead of using Sentence-BERT, we also explore adopting the OpenAI embedding model (text-embedding-ada-002) as the encoder. Specifically, we conduct experiments on 3 datasets: BoolQ, CommonsenseQA and GSM8K. For each dataset, we randomly

---

**Iterative Demonstration Selection**

**Question**: The homeowner frowned at the price of gas, what did he have to do later? Answer Choices: (A) own home (B) mail property tax payments (C) board windows (D) cut grass (E) receive mail
**Iteration 1**: Answer: B\nReason: ...
**Iteration 2**: Answer: D\nReason: ...
**Iteration 3**: Answer: D\nReason: ...

**Label**: D

---

**Top-k-Consistency-CoT**

**Question**: The homeowner frowned at the price of gas, what did he have to do later? Answer Choices: (A) own home (B) mail property tax payments (C) board windows (D) cut grass (E) receive mail
**Response**: Answer: B\nReason: ...; Answer: B\nReason: ...; Answer: B\nReason: ...

**Label**: D

---

**Iterative Demonstration Selection**

**Input**: Texas entrepreneur wants to kick computer gaming up to the next level by offering players a chance at some real-live killing via mouse and modem.

**Iteration 1**
**Examples**:
**Input**: Six days a week, teens crowd the Blue Screen Gaming cybercafe to hunt each other down with assault rifles inside virtual computer worlds...
**Topic**: Technology
...
**Response**: Topic: Technology\nReason: ...

**Iteration 2**: ... **Response**: Topic: Technology ...
**Iteration 3**: ... **Response**: Topic: Technology ...

**Label**: Technology

---

**Random-Voting-CoT**

**Input**: Texas entrepreneur wants to kick computer gaming up to the next level by offering players a chance at some real-live killing via mouse and modem.

**Iteration 1**
**Examples**:
**Input**: The Boston Celtics added a healthy Tom Gugliotta and deleted injured Delonte West. Tom, 34, was activated Wednesday from the injured list after missing seven games ...
**Topic**: Sports
...
**Response**: Topic: Sports\nReason: ...

**Iteration 2**: ... **Response**: Topic: Business ...
**Iteration 3**: ... **Response**: Topic: Sports ...

**Label**: Technology

---

Figure 3: Several case studies. We color correct outputs in green, and wrong outputs in red.

Table 8: Accuracy (%) of different methods with OpenAI embedding model (text-embedding-ada-002) on three datasets.

|  | BoolQ | CommonsenseQA | GSM8K |
|---|---|---|---|
| Top-$k$-Consistency-CoT | 86.0 | 75.4 | 75.8 |
| IDS | **87.2** | **78.0** | **77.6** |

sample 500 test examples and compare IDS with the baseline Top-$k$-Consistency-CoT. The results reported in Table 8 demonstrate IDS's robustness to different embedding models.

**Case Study** To further understand the advantage of IDS, we show several cases in Figure 3. As shown in the upper part of the figure, IDS can iteratively select more diverse demonstration examples than Top-$k$-Consistency-CoT which may be able to correct errors from previous iterations. Compared with Random-Voting-CoT, IDS can find examples that share more similar input-output patterns with the test sample to induce the LLM to generate correct answers (the lower part of the figure).

In addition, we show the ability of IDS to generalize to open-source LLMs and the analysis of average similarity scores in Appendix A.4 ~ A.5, respectively.

## 7 CONCLUSION

In this work, we have introduced Iterative Demonstration Selection (IDS) that can iteratively select examples that are diverse but still strongly correlate with the test sample as demonstrations by leveraging Zero-shot-CoT to improve the performance of in-context learning (ICL). Extensive experimental results and analysis show that IDS can consistently outperform previous ICL demonstration selection baselines.

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

Table 9: Deailed information of different datasets. # refers to 'the number of' and 'full' means the whole set. Note that different random seeds do not result in different samples if the whole set is used.

| | CommonsenseQA | BoolQ | AGNews | DBPedia | SST2 | Amazon |
|---|---|---|---|---|---|---|
| **# Labels** | 5 | 2 | 4 | 14 | 2 | 2 |
| **# Training Samples** | 9741 (full) | 9427 (full) | 10000 | 10000 | 10000 | 10000 |
| **# Test Samples** | 1221 (full) | 2000 | 1000 | 1000 | 872 (full) | 1000 |

Zhuosheng Zhang, Aston Zhang, Mu Li, and Alex Smola. Automatic chain of thought prompting in large language models. In *The Eleventh International Conference on Learning Representations (ICLR 2023)*, 2023a.

Zhuosheng Zhang, Aston Zhang, Mu Li, and Alex Smola. Automatic chain of thought prompting in large language models. In *The Eleventh International Conference on Learning Representations*, 2023b. URL https://openreview.net/forum?id=5NTt8GFjUHkr.

Zhuosheng Zhang, Aston Zhang, Mu Li, Hai Zhao, George Karypis, and Alex Smola. Multimodal chain-of-thought reasoning in language models. *arXiv preprint arXiv:2302.00923*, 2023c.

Zihao Zhao, Eric Wallace, Shi Feng, Dan Klein, and Sameer Singh. Calibrate before use: Improving few-shot performance of language models. In Marina Meila and Tong Zhang (eds.), *Proceedings of the 38th International Conference on Machine Learning*, volume 139 of *Proceedings of Machine Learning Research*, pp. 12697–12706. PMLR, 18–24 Jul 2021. URL https://proceedings.mlr.press/v139/zhao21c.html.

Ce Zheng, Lei Li, Qingxiu Dong, Yuxuan Fan, Zhiyong Wu, Jingjing Xu, and Baobao Chang. Can we edit factual knowledge by in-context learning? *arXiv preprint arXiv:2305.12740*, 2023.

Denny Zhou, Nathanael Schärli, Le Hou, Jason Wei, Nathan Scales, Xuezhi Wang, Dale Schuurmans, Olivier Bousquet, Quoc Le, and Ed Chi. Least-to-most prompting enables complex reasoning in large language models. *arXiv preprint arXiv:2205.10625*, 2022. URL https://arxiv.org/abs/2205.10625.

Yongchao Zhou, Andrei Ioan Muresanu, Ziwen Han, Keiran Paster, Silviu Pitis, Harris Chan, and Jimmy Ba. Large language models are human-level prompt engineers. In *The Eleventh International Conference on Learning Representations*, 2023. URL https://openreview.net/forum?id=92gvk82DE-.

Deyao Zhu, Jun Chen, Xiaoqian Shen, Xiang Li, and Mohamed Elhoseiny. Minigpt-4: Enhancing vision-language understanding with advanced large language models. *arXiv preprint arXiv:2304.10592*, 2023.

# A APPENDIX

## A.1 INSTRUCTIONS AND INPUT FORMATS OF DIFFERENT TASKS

We show the instructions and input formats of different types of tasks for in-context learning in Figure 4.

## A.2 ALGORITHM FOR THE WHOLE SELECTION PROCESS

We illustrate the whole selection process in Algorithm 1

## A.3 DATASETS INFORMATION

We show the detailed information of different datasets in Table 9.

**Sentiment Analysis**

What is the sentiment of the review? Positive or negative?
**Examples**:
**Review**: a captivating drama
**Sentiment**: Positive

...
**Review**: more chaotic than entertaining
**Sentiment**: Negative
The response should follow the format: Sentiment: {positive or negative}\nReason: {reason}
Here is the test data.
**Review**: a tender, heartfelt family drama.
Let's think step by step.

**Topic Classification**

What is the topic of the input? World, sports, business or technology?
**Examples**:
**Input**: Cavs earn fourth straight win...
**Topic**: Sports

...
**Input**: Thomson to Sell Media Unit for $350M...
**Topic**: Business
The response should follow the format: Topic: {world, sports, business or technology}\nReason: {reason}
Here is the test data.
**Input**: Microsoft intros new mice, keyboards...
Let's think step by step.

**Question Answering**

Please answer the question based on the context.
**Examples**:
**Context**: Sikma was voted as one of the...
**Question**: is jack sikma in the hall of fame
**Answer**: Yes

...
**Context**: Timothy Brown is a former football...
**Question**: is tim brown in the hall of fame
**Answer**: Yes
The response should follow the format: Answer: {yes or no}\nReason: {reason}
Here is the test data.
**Context**: Blue is a playful female puppy...
**Question**: is blue off of blue's clues a girl
Let's think step by step.

**Commonsense Reasoning**

Which choice is the correct answer to the question?
**Examples**:
**Question**: If you poke yourself... **Answer Choices**:
(A) have fun...
**Answer**: C

...
**Question**: Where would a bald eagle... **Answer Choices**: (A) great outdoors...
**Answer**: D
The response should follow the format: Answer: {A, B, C, D or E}\nReason: {reason}
Here is the test data.
**Question**: How can I store... **Answer Choices**...
Let's think step by step.

Figure 4: Instructions and input formats of four different categories of tasks (sentiment analysis, topic classification, question answering, and commonsense reasoning) for ICL. For Zero-shot-CoT in the first step of IDS, there is no demonstration example and the instruction "Here is the test data.".

---

**Algorithm 1** Selection process of IDS

---

**Require:** Training set $\mathcal{D}_{\text{train}}$, test set $\mathcal{D}_{\text{test}}$, $\text{LLM}_\theta$, number of demonstrations $k$, number of iterations $q$ and answer set $\hat{A}_{all} = \varnothing$
1: ENCODE all samples in $\mathcal{D}_{\text{train}}$ using Sentence-BERT      ▷ Encode training set
2: **for** $(\hat{x}_i, \hat{y}_i)$ in $\mathcal{D}_{\text{test}}$ **do**
3:     APPLY Zero-shot-CoT to $(\hat{x}_i, \hat{y}_i)$ to obtain the reasoning path R      ▷ Zero-shot-CoT
4:     **for** $j = 1, \dots, q$ **do**
5:        ENCODE R using Sentence-BERT      ▷ Encode reasoning path
6:        USE R to select top-$k$ most similar examples $\mathcal{S} = \{(x_1, y_1), ..., (x_k, y_k)\}$ from $\mathcal{D}_{\text{train}}$ as demonstrations      ▷ KNN selection
7:        $(\hat{A}, \hat{R}) = \text{LLM}_\theta(S, \hat{x}_i)$      ▷ ICL
8:        $R = \hat{R}$, $\hat{A}_{all} = \hat{A}_{all} \cup \{\hat{A}\}$      ▷ Update reasoning path and answer set
9:     **end for**
10:     ADOPT majority voting for $\hat{A}_{all}$ to obtain the final result $\hat{A}_{final}$ for the test sample $(\hat{x}_i, \hat{y}_i)$      ▷ Majority voting
11: **end for**

---

## A.4   GENERALIZATION TO OPEN-SOURCE LLMs

To better verify the generalization ability of IDS, we use vLLM (Kwon et al., 2023) to serve a Llama-2-70b-chat model (Touvron et al., 2023) for experiments and compare IDS with the best

Table 10: Accuracy (%) of different methods with Llama-2-70b-chat.

|  | BoolQ | GSM8K |
|---|---|---|
| Top-$k$-Consistency-CoT | 84.2 | 49.6 |
| IDS | **85.4** | **51.4** |

Table 11: Average similarity scores between test examples and the corresponding selected demonstrations of three methods (Top-$k$-Consistency-CoT, IDS and Random-Voting-CoT).

|  | Top-$k$-Consistency-CoT | IDS | Random-Voting-CoT |
|---|---|---|---|
| Average Similarity Score | 0.69 | 0.46 | 0.31 |

baseline Top-$k$-Consistency-CoT on two datasets: BoolQ and GSM8K. We randomly sample 500 test examples for experiments and report the results in Table 10, which demonstrates that IDS can successfully generalize to open-source LLMs.

## A.5 AVERAGE SIMILARITY SCORES

In Table 11, we report the average similarity scores between test samples and the corresponding demonstrations of different methods. Specifically, we randomly select 200 test examples for each dataset and use Sentence-BERT to obtain contextual representations for calculating similarity scores. We can see that the average similarity score of IDS is between that of Top-$k$-Consistency-CoT and Random-Voting-CoT, indicating that it can indeed strike a balance between two selection dimensions.

