# OpenReview forum: "In-Context Learning with Iterative Demonstration Selection"
_ICLR.cc/2024/Conference — ICLR 2024 Conference Withdrawn Submission_

### Official Review · Reviewer_vsHS · 2023-10-27

**Soundness:** 2 fair
**Presentation:** 3 good
**Contribution:** 2 fair
**Rating:** 3
**Confidence:** 4

**Summary:**

The paper proposes the Iterative Demonstration Selection method to leverage the merits of both dimensions. With Zero-shot-CoT, IDS iteratively selects examples that are diverse but still strongly correlated with the test sample as ICL demonstrations. The proposed method utilizes the output reasoning path to choose demonstrations. After several iterations, IDS adopts majority voting to obtain the final result. The authors assert that the optimal dimension for selecting demonstration examples is task-specific.

**Strengths:**

* The paper is written in a clear manner.
* There is an experiment demonstrating the need to consider diversity and similarity in a task-specific manner.
* There are several analyses regarding the number of demonstrations, the number of iterations, and model types.

**Weaknesses:**

* The experimental results look somewhat marginal.
* The performed experiments are too limited in scope. They were conducted only on CommonsenseQA, BoolQ, AGNews, DBPedia, SST2, and Amazon, which undermines the robustness of the methodology. Results from a more diverse set of tasks are needed e.g., not the classification tasks.
* Measuring cosine similarity between the reasoning path R and the training set lacks some persuasiveness as a criterion for selecting few-shot samples. If there are theoretical or empirical reasons why measuring cosine similarity between reasoning path R and training set samples that have somewhat different forms from reasoning path would be helpful, please provide them.
* There is too much repeated content in the text. e.g."the generated answer is accompanied by its
corresponding reasoning path "
* There is a lack of comparison with other demo selection baselines, not just the internal baseline.

**Questions:**

* How long does it take to measure the correlation between reasoning path R and the whole training set for each task?
* Please provide results using a different encoder model other than Sentence-BERT. It seems necessary to verify if solidity is ensured depending on the encoder.
* Could you provide full results on all tasks, not the average one?

---

> ### Author Response · Authors · 2023-11-18
> **Response to Reviewer vsHS (Part1)**
>
> Dear Reviewer:
>
> We thank you for your thoughtful comments and feedback. We address your questions here.
>
> 1. **The experimental results look somewhat marginal.**
>
> We would like to point out that the average improvement looks somewhat marginal because there are several simple tasks where the baseline results are already pretty high (> 95%). As shown in the following response, IDS can bring an average relative improvement of about 3% on more complex tasks.
>
> 2. **The performed experiments are too limited in scope. They were conducted only on CommonsenseQA, BoolQ, AGNews, DBPedia, SST2, and Amazon, which undermines the robustness of the methodology. Results from a more diverse set of tasks are needed e.g., not the classification tasks.**
>
> Thanks for your suggestion. We further conduct experiments on two more complex datasets: GSM8K (mathematical reasoning) and LogiQA (logical reasoning). Specifically, we randomly sample 500 test examples for experiments and compare IDS with the best baseline Top-k-Consistency-CoT using GPT-3.5 (gpt-3.5-turbo). The results are reported below. We can observe that IDS still achieves better performance than the baseline on more complex tasks.
>
>  |   | GSM8K  | LogiQA |
>  |  ----  | ----  | ----  |
>  | Top-k-Consistency-CoT | 76.2 | 45.4  |
>  | IDS (ours) | 78.4 | 46.8  |
>
> 3. **Measuring cosine similarity between the reasoning path R and the training set lacks some persuasiveness as a criterion for selecting few-shot samples. If there are theoretical or empirical reasons why measuring cosine similarity between reasoning path R and training set samples that have somewhat different forms from reasoning path would be helpful, please provide them.**
>
> Our method is inspired by the finding that the optimal dimension (similarity or diversity) for selecting demonstrations is task-specific (Section 4). Therefore, we hope to incorporate both similarity and diversity into the demonstration selection process. Since cosine similarity is commonly used to measure the similarity between different samples [1,2], we also apply it in our method to select examples that are semantically similar to the test query. To ensure that the selected examples are also diverse enough, instead of using the static test query, we use the generated reasoning path varying in different iterations for selecting different sets of demonstrations.
>
> The conclusion in [3] from ACL 2023 also provides some empirical evidence for our claim. It finds that similarity, diversity, and complexity all contribute to compositional generalization on the test suite it constructs.
>
> [1] What makes good in-context examples for gpt-3?
>
> [2] Complementary explanations for effective in-context learning
>
> [3] How Do In-Context Examples Affect Compositional Generalization?
>
> 4. **There is too much repeated content in the text. e.g."the generated answer is accompanied by its corresponding reasoning path"**
>
> Thanks for your suggestion. We will improve the presentation of our paper and upload a new version as soon as possible.

---

> > ### Author Response · Authors · 2023-11-18
> > **Response to Reviewer vsHS (Part2)**
> >
> > 5. **There is a lack of comparison with other demo selection baselines, not just the internal baseline.**
> >
> > Thanks for your suggestion. We further consider 6 baseline methods:
> >
> > [1] Finding Support Examples for In-Context Learning
> >
> > [2] Fairness-guided Few-shot Prompting for Large Language Models
> >
> > [3] Learning to retrieve prompts for in-context learning
> >
> > [4] Synchromesh: Reliable code generation from pre-trained language models
> >
> > [5] Compositional exemplars for in-context learning
> >
> > [6] Skill-Based Few-Shot Selection for In-Context Learning
> >
> > However, [1,2,3,5] require the detailed output distributions of language models, which are not available for gpt-3.5-turbo or gpt-4. And the method to select relevant examples proposed by [4] can only be applied to code generation tasks. In addition, as shown in [6], Skill-KNN (the method in [6]) can achieve better performance than [3,4,5]. Therefore, we further compare our method with Skill-KNN on three datasets: GSM8K, BoolQ and CommonsenseQA.
> >
> > Specifically, we randomly select 8 examples from the training set as seed data and annotate them by querying gpt-3.5-turbo with the prompt ‘What skills do we need to solve this task?’. These 8 examples will serve as in-context learning demonstrations for annotating other examples in the training set and test samples. After annotating the skills, we use the same method as Skill-KNN to select similar examples.
> >
> > Besides, we also try G-fair-Prompting in [2] with Llama-2-13b for selection. The selected examples are directly used as ICL demonstrations for querying gpt-3.5-turbo.
> >
> > For BoolQ and CommonsenseQA, we use the same number of test data as Table 9. For GSM8K, we randomly select 500 test samples for experiments. The results of different methods are shown below. We can see that IDS consistently outperforms new baselines.
> >
> > |   | GSM8K  | BoolQ | CommonsenseQA  |
> > |  ----  | ----  | ----  | ----  |
> > | G-fair-Prompting | 75.2 | 84.8 | 75.5 |
> > | Skill-KNN | 77.0 | 85.9  | 75.2  |
> > | IDS (ours) | 78.4 | 87.8 | 78.1 |
> >
> > 6. **How long does it take to measure the correlation between reasoning path R and the whole training set for each task?**
> >
> > As we use Faiss to index the embeddings of training examples, measuring the correlation between reasoning path R and the whole training set takes negligible time compared to querying the OpenAI API. For each dataset, we use the same number of test examples as Table 9 and calculate the total time for measuring the correlation. The results shown below verify that the time is negligible.
> >
> > |   | CommonsenseQA  | BoolQ | AGNews  | DBPedia | SST2 | Amazon |
> > |  ----  | ----  | ----  | ----  |  ----  | ----  | ----  |
> > | Time (s) | 2.4 | 3.3 | 1.7 | 1.9 | 1.5 | 1.7 |
> >
> > 7. **Please provide results using a different encoder model other than Sentence-BERT. It seems necessary to verify if solidity is ensured depending on the encoder.**
> >
> > Thanks for your suggestion. We further conduct experiments with the OpenAI embedding model (text-embedding-ada-002) on three datasets: BoolQ, CommonsenseQA and GSM8K. For each dataset, we randomly sample 500 test examples for experiments and compare IDS with the best baseline Top-k-Consistency-CoT using GPT-3.5 (gpt-3.5-turbo). From the results shown below, we can observe that IDS still performs better than the baseline when using the OpenAI embedding model for retrieval.
> >
> > |   | BoolQ  | CommonsenseQA | GSM8K  |
> > |  ----  | ----  | ----  | ----  |
> > | Top-k-Consistency-CoT | 86.0 | 75.4  | 75.8  |
> > | IDS (ours) | 87.2 | 78.0 | 77.6 |
> >
> > 8. **Could you provide full results on all tasks, not the average one?**
> >
> > We believe there is a misunderstanding regarding the reporting of results. As shown in Table 3, we have provided detailed results for each dataset.
> >
> > Thanks again for your feedback. We hope that this addresses your questions.

---

> > > ### Comment · Reviewer_vsHS · 2023-11-21
> > > **Reply to the response**
> > >
> > > Thank you for your rebuttal.
> > >
> > > Firstly, I did not solely base my comments on average accuracy. As seen in Table 3, not only for tasks where the baselines exceed 95% but also for other tasks, the performance gaps still seem marginal.
> > >
> > > Secondly, I appreciate the additional validation on two more tasks. However, why were only Top-k-Consistency comparisons made for these two tasks? Is there a specific reason for this?
> > >
> > > Regarding the point that cosine similarity is commonly used to measure similarity, I acknowledge this, but my question was specifically about measuring the similarity between "demo" and "reasoning path" and choosing demos based on that.
> > >
> > > As for my request for results on all tasks, I was seeking results for the remaining tables (e.g., Table 4, ...) other than Table 3 as they present averages. Additionally, in relation to experiments mentioning 'randomly choose one seed for experiments,' is there a particular reason for this?
> > >
> > > While I appreciate the authors' rebuttal, I will maintain my rating for now.

---

> > > > ### Author Response · Authors · 2023-11-22
> > > > **Response to Reviewer vsHS**
> > > >
> > > > Dear Reviewer:
> > > >
> > > > Thank you for taking the time to read our rebuttal. We address your questions here.
> > > >
> > > > 1. **Firstly, I did not solely base my comments on average accuracy. As seen in Table 3, not only for tasks where the baselines exceed 95% but also for other tasks, the performance gaps still seem marginal.**
> > > >
> > > > If we remove two tasks where the baselines exceed 95% (DBPedia and Amazon) and add two more complex tasks (GSM8K and LogiQA), the average performance of IDS is 79.5 while the average performance of the best baseline Top-k-Consistency-CoT is 77.9. We can see that IDS brings an average performance improvement of 1.6%, which is not that marginal considering that the increase in API requests for IDS is negligible.
> > > >
> > > >
> > > > 2. **Secondly, I appreciate the additional validation on two more tasks. However, why were only Top-k-Consistency comparisons made for these two tasks? Is there a specific reason for this?**
> > > >
> > > > We did not have enough time and budget to evaluate all methods on all investigated datasets during the rebuttal. As Top-k-Consistency-CoT is the best baseline method in Table 3, we conduct experiments with it for comparison.
> > > >
> > > >
> > > > 3. **Regarding the point that cosine similarity is commonly used to measure similarity, I acknowledge this, but my question was specifically about measuring the similarity between "demo" and "reasoning path" and choosing demos based on that.**
> > > >
> > > > If we use the test query to select demonstrations, we can only obtain similar examples (**similarity**).
> > > >
> > > > To incorporate diversity into the demonstration selection process, instead of using the static test query, we use the generated reasoning path varying in different iterations for selecting different sets of demonstrations (**diversity**). Besides, as the generated reasoning path is strongly correlated with the original test query, the selected demonstrations can also ensure **similarity**.
> > > >
> > > >
> > > > 4. **As for my request for results on all tasks, I was seeking results for the remaining tables (e.g., Table 4, ...) other than Table 3 as they present averages. Additionally, in relation to experiments mentioning 'randomly choose one seed for experiments,' is there a particular reason for this?**
> > > >
> > > > Due to the cost of API requests, we decided to randomly choose one seed for experiments.
> > > >
> > > > There is not enough space to report the full results for all tables. We will include the full results in the appendix.
> > > >
> > > > Thanks again for your feedback. We hope that this addresses your questions.

---

### Official Review · Reviewer_R6Td · 2023-10-29

**Soundness:** 3 good
**Presentation:** 3 good
**Contribution:** 2 fair
**Rating:** 6
**Confidence:** 5

**Summary:**

The paper addresses the challenge of selecting the most suitable few-shot demonstrations for in-context learning (ICL) in large language models (LLMs).
The authors argue that the optimal selection dimension, i.e., diversity or similarity, is task-specific and propose an Iterative Demonstration Selection (IDS) method that leverages the merits of both dimensions.
IDS uses zero-shot chain-of-thought reasoning (Zero-shot-CoT) to iteratively select examples that are diverse but still strongly correlated with the test sample as ICL demonstrations.
Experiments on various tasks demonstrate the effectiveness of IDS.

**Strengths:**

1. The methodology is well-explained, with IDS applying Zero-shot-CoT to the test sample before demonstration selection. The output reasoning path is iteratively used to choose demonstrations that are prepended to the test sample for inference. After several iterations, IDS adopts majority voting to obtain the final result.

2. Experiments on various tasks and thorough analysis on hyper-parameters (e.g., number of demonstrations and number of iterations) demonstrate the effectiveness of IDS.

**Weaknesses:**

1. Lack of comparison with stronger baselines. Much related work and methods for in-context example selection (e.g., EPR [1], TST[2], CEIL[3], Skill-KNN[4]) are not experimentally compared (or even not mentioned). At least some of them should appear in the experiments part.

2. Actually, this work is not "the first time consider both the diversity and similarity dimensions of ICL demonstration selection for LLMs".
For instance, [5] use MMR that considers both similarity and diversity. [6] also demonstrates the effectivenss of incorporating both similarity and diversity.

[1] Ohad Rubin, Jonathan Herzig, and Jonathan Berant. "Learning to retrieve prompts for in-context learning."

[2] Gabriel Poesia, et al. "Synchromesh: Reliable code generation from pre-trained language models."

[3] J. Ye, et al. "Compositional exemplars for in-context learning."

[4] S. An, et al. "Skill-Based Few-Shot Selection for In-Context Learning."

[5] X. Ye, et al. "Complementary explanations for effective in-context learning."

[6] S. An, et al. "How Do In-Context Examples Affect Compositional Generalization?."

**Questions:**

How would IDS perform without voting, i.e., just use the final answer achieved by the last iteration? I suppose this result better reflect the effectiveness of the designed iteration.

---

> ### Author Response · Authors · 2023-11-18
> **Response to Reviewer R6Td**
>
> Dear Reviewer:
>
> We thank you for your thoughtful comments and feedback. We address your questions here.
>
> 1. **Lack of comparison with stronger baselines. Much related work and methods for in-context example selection (e.g., EPR, TST, CEIL, Skill-KNN) are not experimentally compared (or even not mentioned). At least some of them should appear in the experiments part.**
>
> Thanks for your suggestion. We further consider 6 baseline methods:
>
> [1] Finding Support Examples for In-Context Learning
>
> [2] Fairness-guided Few-shot Prompting for Large Language Models
>
> [3] Learning to retrieve prompts for in-context learning
>
> [4] Synchromesh: Reliable code generation from pre-trained language models
>
> [5] Compositional exemplars for in-context learning
>
> [6] Skill-Based Few-Shot Selection for In-Context Learning
>
> However, [1,2,3,5] require the detailed output distributions of language models, which are not available for gpt-3.5-turbo or gpt-4. And the method to select relevant examples proposed by [4] can only be applied to code generation tasks. In addition, as shown in [6], Skill-KNN (the method in [6]) can achieve better performance than [3,4,5]. Therefore, we further compare our method with Skill-KNN on three datasets: GSM8K, BoolQ and CommonsenseQA.
>
> Specifically, we randomly select 8 examples from the training set as seed data and annotate them by querying gpt-3.5-turbo with the prompt ‘What skills do we need to solve this task?’. These 8 examples will serve as in-context learning demonstrations for annotating other examples in the training set and test samples. After annotating the skills, we use the same method as Skill-KNN to select similar examples.
>
> Besides, we also try G-fair-Prompting in [2] with Llama-2-13b for selection. The selected examples are directly used as ICL demonstrations for querying gpt-3.5-turbo.
>
> For BoolQ and CommonsenseQA, we use the same number of test data as Table 9. For GSM8K, we randomly select 500 test samples for experiments. The results of different methods are shown below. We can see that IDS consistently outperforms new baselines.
>
> |   | GSM8K  | BoolQ | CommonsenseQA  |
> |  ----  | ----  | ----  | ----  |
> | G-fair-Prompting | 75.2 | 84.8 | 75.5 |
> | Skill-KNN | 77.0 | 85.9  | 75.2  |
> | IDS (ours) | 78.4 | 87.8 | 78.1 |
>
> 2. **Actually, this work is not "the first time consider both the diversity and similarity dimensions of ICL demonstration selection for LLMs". For instance, [7] use MMR that considers both similarity and diversity. [8] also demonstrates the effectivenss of incorporating both similarity and diversity.**
>
> [7] X. Ye, et al. "Complementary explanations for effective in-context learning."
>
> [8] S. An, et al. "How Do In-Context Examples Affect Compositional Generalization?."
>
> Thanks for providing these two papers. We were not aware of them from ACL-2023 before.  We will cite them and revise the claim in our paper accordingly.
>
>
> 3. **How would IDS perform without voting, i.e., just use the final answer achieved by the last iteration? I suppose this result better reflect the effectiveness of the designed iteration.**
>
> Thanks for your suggestion. However, according to our understanding, using the final answer obtained in the last iteration can not reflect the effectiveness of our method as we use self-consistency / majority voting in baseline methods. For IDS without voting, we analyze the performance of IDS and baselines with different numbers of iterations q in Section 6.3 which demonstrates that IDS can still outperform baselines even without voting (q = 1).
>
>
> Thanks again for your feedback. We hope that this addresses your questions.

---

> > ### Comment · Reviewer_R6Td · 2023-11-22
> > **Thanks for your response**
> >
> > The additional experiments slightly improves the soundness of this work. I update my score.

---

### Official Review · Reviewer_LAeZ · 2023-10-30

**Soundness:** 3 good
**Presentation:** 2 fair
**Contribution:** 2 fair
**Rating:** 5
**Confidence:** 4

**Summary:**

This paper proposes Iterative Demonstration Selection (IDS) to conjoin diversity and similarity for tackling the problem of demonstration selection for in-context learning of LLMs. Technically, IDS leverages intermediate reasoning paths provided by LLMs to retrieve relevant training samples. IDS is evaluated on several representative tasks.

**Strengths:**

- The paper is well motivated with preliminary experimental results.
- The literature review is good.

**Weaknesses:**

- Overall, the presentation of the paper should be improved, and the technical novelty is limited. In particular, Figure 2 should be carefully polished.
- The explanation of why IDS can incorporate diversity should be made clear. I notice the argument "they can be
different during iterations to ensure diversity because the reasoning paths vary in different iterations", but how can you ensure such diversity? Purely rely on the randomness in LLM sampling?
- It seems that the evaluation tasks are relatively simple. Have you tested IDS on GSM8K or MATH?
- In my opinion, when IDS and the baselines use the same number of reasoning paths or iterations, IDS actually uses one more query to GPT than the baselines (due to the first zero-shot CoT). As a result, in the 1-iteration column of Table 6, IDS is better. If so, I want to know if you give the baselines one more query to GPT, can their performance be further improved?
- IDS's performance should heavily rely on the metrics used for retrieval. Have you tested other choices except for Bert Distance?

**Questions:**

See above

---

> ### Author Response · Authors · 2023-11-18
> **Response to Reviewer LAeZ**
>
> Dear Reviewer:
>
> We thank you for your thoughtful comments and feedback. We address your questions here.
>
> 1. **Overall, the presentation of the paper should be improved, and the technical novelty is limited. In particular, Figure 2 should be carefully polished.**
>
> Thanks for your suggestion. We will improve the presentation of our paper and upload a new version as soon as possible.
>
> We would like to point out that our technical novelty lies mainly in considering both the diversity and similarity dimensions of ICL demonstration selection for LLMs. We identify that the optimal dimension for selecting demonstrations is task-specific and propose Iterative Demonstration Selection to fully leverage the merits of both dimensions.
>
> 2. **The explanation of why IDS can incorporate diversity should be made clear. I notice the argument "they can be different during iterations to ensure diversity because the reasoning paths vary in different iterations", but how can you ensure such diversity? Purely rely on the randomness in LLM sampling?**
>
> As indicated in Section 5, diversity is realized by different inputs and LLM sampling.
>
> For the first iteration, the demonstrations in the input D1 are selected by the reasoning path R0 generated in Zero-shot-Cot. D1 is then used for ICL to generate R1. Note that R1 is likely to be different from R0 due to different inputs and LLM sampling.
>
> For iteration 2, the demonstrations in the input D2 are selected by the reasoning path R1 generated in iteration 1. As R1 is different from R0 (mentioned above), D2 is likely to be different from D1, which will contribute to the difference between R2 and R1.
>
> Subsequent iterations are the same.
>
> Therefore, diversity can be achieved by different inputs and LLM sampling.
>
> 3. **It seems that the evaluation tasks are relatively simple. Have you tested IDS on GSM8K or MATH?**
>
> Thanks for your suggestion. We further conduct experiments on two more complex datasets: GSM8K (mathematical reasoning) and LogiQA (logical reasoning). Specifically, we randomly sample 500 test examples for experiments and compare IDS with the best baseline Top-k-Consistency-CoT using GPT-3.5 (gpt-3.5-turbo). The results are reported below. We can observe that IDS still achieves better performance than the baseline on more complex tasks.
>
>  |   | GSM8K  | LogiQA |
>  |  ----  | ----  | ----  |
>  | Top-k-Consistency-CoT | 76.2 | 45.4  |
>  | IDS (ours) | 78.4 | 46.8  |
>
> 4. **In my opinion, when IDS and the baselines use the same number of reasoning paths or iterations, IDS actually uses one more query to GPT than the baselines (due to the first zero-shot CoT). As a result, in the 1-iteration column of Table 6, IDS is better. If so, I want to know if you give the baselines one more query to GPT, can their performance be further improved?**
>
> Actually, the increase in API request cost of Zero-shot-CoT in the first step is negligible as the number of tokens in Zero-shot-CoT is much less than that in ICL. Following your suggestion, we also add one Zero-shot-CoT query to the baseline. If the result of Zero-shot-CoT is different from that of ICL, we randomly select one as the final result. However, the performance becomes even worse (89.8 -> 89.0).
>
>
> 5. **IDS's performance should heavily rely on the metrics used for retrieval. Have you tested other choices except for Bert Distance?**
>
> Thanks for your suggestion. We further conduct experiments with the OpenAI embedding model (text-embedding-ada-002) on three datasets: BoolQ, CommonsenseQA and GSM8K. For each dataset, we randomly sample 500 test examples for experiments and compare IDS with the best baseline Top-k-Consistency-CoT using GPT-3.5 (gpt-3.5-turbo). From the results shown below, we can see that IDS still performs better than the baseline when using the OpenAI embedding model for retrieval.
>
> |   | BoolQ  | CommonsenseQA | GSM8K  |
> |  ----  | ----  | ----  | ----  |
> | Top-k-Consistency-CoT | 86.0 | 75.4  | 75.8  |
> | IDS (ours) | 87.2 | 78.0 | 77.6 |
>
> Thanks again for your feedback. We hope that this addresses your questions.

---

> > ### Comment · Reviewer_LAeZ · 2023-11-22
> > **Thanks**
> >
> > Thanks for the rebuttal, which has actually addressed some of my concerns. However, I still hold concerns about the mentioned diversity, which stems from the unmanageable randomness in LLM inputs and the sampling process. If so, you should carefully revise the arguments regarding ensuring diversity in the manuscript, or develop techniques that can ensure diversity in a more reliable way.

---

> > > ### Author Response · Authors · 2023-11-22
> > > **Response to Reviewer LAeZ**
> > >
> > > Thank you for taking the time to read our rebuttal. We will revise the arguments regarding ensuring diversity accordingly.

---

### Official Review · Reviewer_MpmC · 2023-10-31

**Soundness:** 2 fair
**Presentation:** 3 good
**Contribution:** 2 fair
**Rating:** 5
**Confidence:** 4

**Summary:**

This paper proposes a strategy, Iterative Demonstration Selection (IDS), used for example selection in in-context learning (ICL) setting of large language models (LLMs), focusing on the balance between diversity and similarity. IDS leverages the reasoning path elicited by zero-shot chain-of-thought (CoT) for similarity between the test sample and demonstrations and iterative selection for diversity among ICL examples. They evaluate IDS on several NLP datasets (commonsense reasoning, question answering, topic classification, and sentiment analysis) and show it consistently outperforms existing ICL demonstration selection methods.

**Strengths:**

1. The paper is well-written, and the proposed method IDS is easy to follow.

2. This paper provides a conclusion that both similarity and diversity are important in example selection in ICL scenarios, which can help other researchers who are working on a similar area.

**Weaknesses:**

1. Experiments are conducted based on simple tasks such as classification, commonsense reasoning, etc., on which the improvements seem marginal. More complex and difficult generative tasks, including mathematical reasoning, QA, and machine translation, are encouraged to be adapted.
2. The conclusion that “it is unreasonable to claim that one dimension is consistently better than the other across different tasks” is drawn through only two datasets, AGNews and CommonsenseQA, which is not that solid.
3. IDS based on voting brings in 4x of the overhead for API requests, while the improvements are a little marginal.
4. Experiments are conducted on GPT-3.5-turbo. More LLMs, such as Vicuna, Llama, and Alpaca, can be tested.

**Questions:**

1. Comparison with other example selection strategies has yet to be explored. Does the balance matter? The exploration for similarity and diversity is only conducted on 2 datasets.


[1] Li, Xiaonan, and Xipeng Qiu. "Finding supporting examples for in-context learning." arXiv preprint arXiv:2302.13539 (2023).
[2] Ma, Huan, et al. "Fairness-guided Few-shot Prompting for Large Language Models." arXiv preprint arXiv:2303.13217 (2023).

---

> ### Author Response · Authors · 2023-11-18
> **Response to Reviewer MpmC (Part1)**
>
> Dear Reviewer:
>
> We thank you for your thoughtful comments and feedback. We address your questions here.
>
> 1. **Experiments are conducted based on simple tasks such as classification, commonsense reasoning, etc., on which the improvements seem marginal. More complex and difficult generative tasks, including mathematical reasoning, QA, and machine translation, are encouraged to be adapted.**
>
> Thanks for your suggestion. We further conduct experiments on two more complex datasets: GSM8K (mathematical reasoning) and LogiQA (logical reasoning). Specifically, we randomly sample 500 test examples for experiments and compare IDS with the best baseline Top-k-Consistency-CoT using GPT-3.5 (gpt-3.5-turbo). The results are reported below. We can observe that IDS still achieves better performance than the baseline on more complex tasks.
>
>  |   | GSM8K  | LogiQA |
>  |  ----  | ----  | ----  |
>  | Top-k-Consistency-CoT | 76.2 | 45.4  |
>  | IDS (ours) | 78.4 | 46.8  |
>
> 2. **The conclusion that “it is unreasonable to claim that one dimension is consistently better than the other across different tasks” is drawn through only two datasets, AGNews and CommonsenseQA, which is not that solid. / Does the balance matter? The exploration for similarity and diversity is only conducted on 2 datasets.**
>
> We would like to point out that except for the pilot experiments on AGNews and CommonsenseQA in Section 4, the main experiments covering 6 different datasets in Section 6 (Table 3) also support the conclusion that the optimal dimension for selecting demonstration examples is task-specific.
>
> To better support our claim, we add some pilot experiments on BoolQ and SST2. Specifically, we randomly sample 100 examples from the original test set for experiments and conduct 4-shot learning using GPT-3.5 (gpt-3.5-turbo). The results shown below also verify the conclusion.
>
>  |   | BoolQ  | SST2 |
>  |  ----  | ----  | ----  |
>  | Similar-ICL-Consistency (Similarity) | 84.8 | 94.3  |
>  | Random-ICL-Voting (Diversity) | 83.5 | 95.2  |
>
> 3. **IDS based on voting brings in 4x of the overhead for API requests, while the improvements are a little marginal.**
>
> We believe there is a misunderstanding about the overhead for API requests. Actually, the increase in API requests for IDS is negligible:
>
> (1) As mentioned in Section 6.1, for the similarity-based method, we apply self-consistency with **q** decoding paths to match the number of iterations **q**. For the diversity-based method, we run experiments **q** times before majority voting to match the number of iterations **q**.
>
> (2) To ensure that the cost per API request is the same for IDS and baselines, we also apply Zero-shot-CoT to baseline methods to allow them to simultaneously generate answers and reasoning paths.
>
> Therefore, IDS can consistently outperform baseline methods on all datasets with a negligible increase in API request cost (Zero-shot-CoT in the first step) as the number of tokens in Zero-shot-CoT is much less than that in ICL.
>
> In addition, we also conduct controlled experiments with different numbers of iterations in Section 6.3 which demonstrates that IDS can still outperform baselines even without iterations (q = 1).
>
> 4. **Experiments are conducted on GPT-3.5-turbo. More LLMs, such as Vicuna, Llama, and Alpaca, can be tested.**
>
> As explained in the introduction, we mainly focus on LLMs for which parameters or detailed output distributions are usually not available. Therefore, we explore **gpt-3.5-turbo and gpt-4** in our work.
>
> To better demonstrate the generalization ability of our method, we use vLLM to serve a **Llama-2-70b-chat** model for experiments. Specifically, we compare IDS with the best baseline Top-k-Consistency-CoT on two datasets: BoolQ and GSM8K. For each dataset,  we randomly sample 500 test examples for experiments. The results shown below also verify the generalization capability of IDS.
>
>  |   | BoolQ  | GSM8K |
>  |  ----  | ----  | ----  |
>  | Top-k-Consistency-CoT | 84.2 | 49.6  |
>  | IDS (ours) | 85.4 | 51.4 |

---

> > ### Author Response · Authors · 2023-11-18
> > **Response to Reviewer MpmC (Part2)**
> >
> > 5. **Comparison with other example selection strategies has yet to be explored.**
> >
> > Thanks for your suggestion. We further consider 6 baseline methods:
> >
> > [1] Finding Support Examples for In-Context Learning
> >
> > [2] Fairness-guided Few-shot Prompting for Large Language Models
> >
> > [3] Learning to retrieve prompts for in-context learning
> >
> > [4] Synchromesh: Reliable code generation from pre-trained language models
> >
> > [5] Compositional exemplars for in-context learning
> >
> > [6] Skill-Based Few-Shot Selection for In-Context Learning
> >
> > However, [1,2,3,5] require the detailed output distributions of language models, which are not available for gpt-3.5-turbo or gpt-4. And the method to select relevant examples proposed by [4] can only be applied to code generation tasks. In addition, as shown in [6], Skill-KNN (the method in [6]) can achieve better performance than [3,4,5]. Therefore, we further compare our method with Skill-KNN on three datasets: GSM8K, BoolQ and CommonsenseQA.
> >
> > Specifically, we randomly select 8 examples from the training set as seed data and annotate them by querying gpt-3.5-turbo with the prompt ‘What skills do we need to solve this task?’. These 8 examples will serve as in-context learning demonstrations for annotating other examples in the training set and test samples. After annotating the skills, we use the same method as Skill-KNN to select similar examples.
> >
> > Besides, we also try G-fair-Prompting in [2] with Llama-2-13b for selection. The selected examples are directly used as ICL demonstrations for querying gpt-3.5-turbo.
> >
> > For BoolQ and CommonsenseQA, we use the same number of test data as Table 9. For GSM8K, we randomly select 500 test samples for experiments. The results of different methods are shown below. We can see that IDS consistently outperforms new baselines.
> >
> > |   | GSM8K  | BoolQ | CommonsenseQA  |
> > |  ----  | ----  | ----  | ----  |
> > | G-fair-Prompting | 75.2 | 84.8 | 75.5 |
> > | Skill-KNN | 77.0 | 85.9  | 75.2  |
> >  | IDS (ours) | 78.4 | 87.8 | 78.1 |
> >
> > Thanks again for your feedback. We hope that this addresses your questions.

---

> > > ### Comment · Reviewer_MpmC · 2023-11-22
> > > **Thanks for the response**
> > >
> > > Thanks for the Rebuttal. After carefully reading the discussion as well as other reviewers' comments. I want to keep my score.

---

### Author Response · Authors · 2023-11-18
**General response to all reviewers**

We appreciate the valuable comments of all reviewers. We have uploaded a revised version of our paper that incorporates the suggestions in the reviews. And the changes are marked in blue. We believe the new version is clearer based on the valuable suggestions. Please let us know if our responses address your questions or concerns. We will be very happy to respond to further questions. Thank you all once again.